# Exploring the CNOT1(800–999) HEAT Domain and Its Interactions with Tristetraprolin (TTP) as Revealed by Hydrogen/Deuterium Exchange Mass Spectrometry

**DOI:** 10.3390/biom15030403

**Published:** 2025-03-11

**Authors:** Maja K. Cieplak-Rotowska, Michał Dadlez, Anna Niedzwiecka

**Affiliations:** 1Division of Biophysics, Institute of Experimental Physics, Faculty of Physics, University of Warsaw, PL-02089 Warsaw, Poland; majacieplak@gmail.com; 2Laboratory of Biological Physics, Institute of Physics, Polish Academy of Sciences, Aleja Lotnikow 32/46, PL-02668 Warsaw, Poland; 3Laboratory of Mass Spectrometry, Institute of Biochemistry and Biophysics, Polish Academy of Sciences, PL-02106 Warsaw, Poland; michald@ibb.waw.pl

**Keywords:** CNOT1, HEAT repeat, TTP, hydrogen–deuterium exchange, mass spectrometry, protein conformational dynamics

## Abstract

CNOT1, a key scaffold in the CCR4-NOT complex, plays a critical role in mRNA decay, particularly in the regulation of inflammatory responses through its interaction with tristetraprolin. A fragment of the middle part of CNOT1 (residues 800–999) is an example of an α-helical HEAT-like repeat domain. The HEAT motif is an evolutionarily conserved motif present in scaffolding and transport proteins across a wide range of organisms. Using hydrogen/deuterium exchange mass spectrometry (HDX MS), a method that has not been widely explored in the context of HEAT repeats, we analysed the structural dynamics of wild-type CNOT1(800–999) and its two double point mutants (E893A/Y900A, E893Q/Y900H) to find the individual contributions of these CNOT1 residues to the molecular recognition of tristetraprolin (TTP). Our results show that the differences in the interactions of CNOT1(800–999) variants with the TTP peptide fragment are due to the absence of the critical residues resulting from point mutations and not due to the perturbation of the protein structure. Nevertheless, the HDX MS was able to detect slight local changes in structural dynamics induced by protein point mutations, which are usually neglected in studies of intermolecular interactions.

## 1. Introduction

A considerable number of recent studies have demonstrated that protein structure stability and structural flexibility or fluctuations are as important for the function of biological molecules as the protein sequence itself and its averaged spatial structure, e.g., [1,2,3,4]. In recent years, a significant number of biologically relevant questions related to the conformational dynamics and allosteric regulation of proteins have been addressed using a combination of hydrogen–deuterium exchange and mass spectrometry (HDX MS), e.g., [5,6,7,8,9,10,11,12,13,14], also reviewed in: [15,16]. Hydrogen–deuterium exchange coupled with mass spectrometry HDX MS provides a relatively detailed insight into the properties of proteins, since this technique allows the local dynamics of the structural elements of the protein molecule to be mapped. This is possible because the stability of hydrogen bonding networks is reflected in the kinetics of the hydrogen–deuterium exchange of main chain amide protons [17,18,19,20].

This means that HDX MS can provide information on protein flexibility, internal motion, and overall structure by indicating flexible regions vs. regions entangled in more or less stable substructures [21,22,23], which can help to identify conditions in which proteins are stable, the location of binding sites, and changes in structural dynamics associated with ligand binding. In principle, HDX can be measured using any method that will distinguish the two different hydrogen isotopes from each other, e.g., NMR [24,25], MS [26,27], or infrared spectroscopy [28].

Novel discoveries in protein function and activity often arise from observing changes due to point mutations. Most studies focus on demonstrating the functional consequences of these mutations, such as their impact on binding affinity or enzymatic activity, typically assuming they do not affect the overall protein structure. However, proving this assumption is challenging. HDX MS addresses this by allowing the simultaneous study of both protein structure and interactions at peptide-length resolution.

HDX MS is also more widely applicable than multidimensional nuclear magnetic resonance (NMR), since it is more suitable for large proteins and homo-oligomers (e.g., [29]), does not require such high concentrations and long-lasting experiments, and isotopic labelling is performed in the protein solution immediately before measurement. This reduces the limitations associated with low protein overexpression, proteolytic degradation, aggregation, and precipitation.

CNOT1 is a large, α-helical, non-enzymatic protein factor involved in building the main eukaryotic CCR4-NOT deadenylase complex [30]. The Carbon Catabolite repression 4 (CCR4)-negative on TATA-less (NOT) complex is engaged in many different processes, among them: mRNA 3′ deadenylation, histone methylation, DNA repair, transcriptional regulation [31], nuclear RNA surveillance, RNA export, protein modification, protein quality control and cell growth [32,33,34]. The complex possesses two enzymatic activities: nucleolytic mRNA deadenylation, which is catalysed by two different subunits CNOT6/CNOT6L [35,36,37], and CNOT7/CNOT8 [38], and ubiquitination, catalysed by CNOT4. CNOT1 is the largest protein of all of the subunits and serves as a scaffold to which all the other subunits are anchored, either directly or indirectly [39]. Although this protein does not contain any catalytic domains, it is indispensable for deadenylation, since depleting CNOT1 destabilizes the complex [40]. CNOT1 is also where most of the RNA-binding proteins that determine the target of CCR4-NOT bind to, e.g., tristetraprolin (TTP) [41,42], mammalian Nanos [43], and GW182 paralogues [44,45,46] (Figure 1).

The CNOT1 function is fundamental to two gene silencing regulatory pathways mediated by sequences encoded in the 3′-untranslated regions (3′UTRs) of the target mRNAs, i.e., the microRNA (miRNA) binding sites and the adenine/uridine (AU)-rich elements (Figure 1).

In the miRNA-dependent gene silencing pathway, the translation of targeted mRNAs is first repressed [47,48,49,50,51,52,53,54], due to, most likely, inhibition of the mRNA 5′ cap-dependent translation initiation. After inhibition of translation, [47,49,50,51] mRNAs undergo deadenylation mainly by the CCR4-NOT complex [55,56,57], to which they are indirectly recruited by the GW182 protein via its concurrent interactions with PABP, CNOT1 [44,45,46] and CNOT9 [58,59]. This is followed by decapping by DCP1-DCP2 [57,60,61] and further decay by XRN1 [61].

**Figure 1 biomolecules-15-00403-f001:**
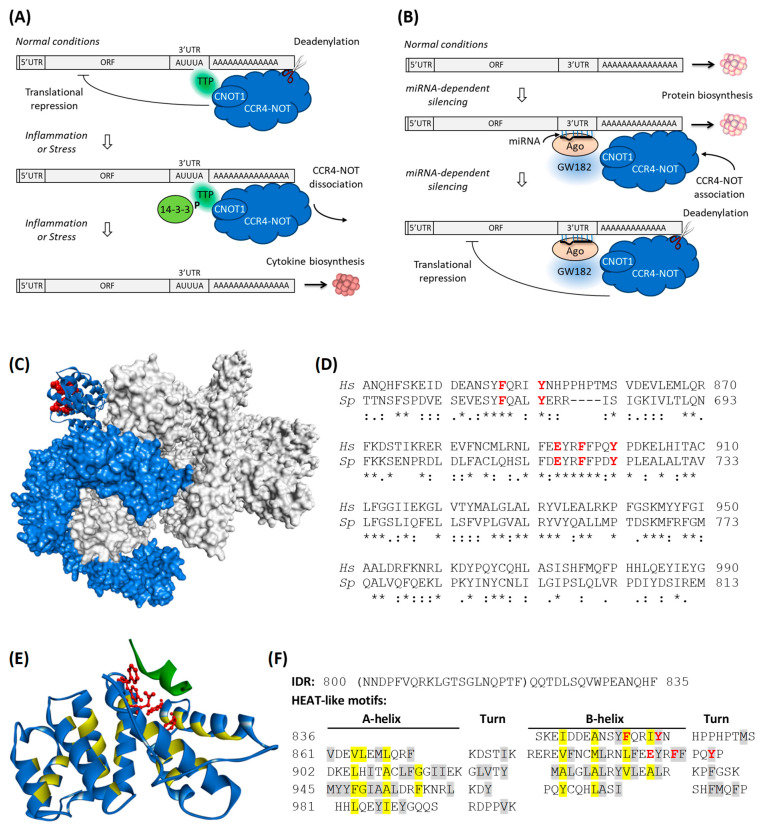
(**A**) Schematic of the ARE-mediated silencing of cytokine genes. Tristetraprolin (TTP) binds to AU-rich elements in the 3′ UTR of cytokine mRNA via its N-terminal zinc fingers and recruits the CCR4-NOT complex via its C-terminal α-helix, which interacts with the scaffolding CCR4-NOT subunit, CNOT1. This interaction leads to deadenylation of the 3′ poly (**A**) tail and translational repression at the 5′ end, thereby inhibiting cytokine production. Under conditions of inflammation or stress, TTP is phosphorylated (P) by 14-3-3 proteins, leading to dissociation of the CCR4-NOT complex from the mRNA and triggering cytokine biosynthesis. (**B**) Schematic of the miRNA-mediated gene silencing. Protein biosynthesis is controlled by the imperfectly complementary binding of miRNA to the 3′ UTR of the target mRNA, along with an Argonaute (Ago) protein and the GW182 protein, that form together the miRNA-induced silencing complex (miRISC) [62,63]. The GW182 protein binds to Ago via its N-terminal domain, while using its C-terminal silencing domain, it recruits the CCR4-NOT complex through its scaffolding subunit, CNOT1, triggering 3′ poly (**A**) tail deadenylation and translation repression at the 5′ end, thereby inhibiting protein production. (**C**) The solvent-accessible surface of the cryo-EM 3D reconstruction of the *S. pombe* CCR4-NOT complex [64,65,66], including a fragment of CNOT1 (blue). The CNOT1 region homologous to human CNOT1(800–999) is shown as a ribbon. The side chains of the residues belonging to the TTP binding site are shown in red in the CPK representation. (**D**) Sequence alignment and homology of the TTP-binding domain of CNOT1 from *H. sapiens* (*Hs*) and *S. pombe* (*Sp*). Identical conserved residues are marked as “ * ”, strongly similar residues as “ : ” and residues that have weakly similar properties as “ . ”. Key residues involved in TTP binding are shown in bold red. (**E**) Crystal structure of the human CNOT1(820–999) fragment (blue) in complex with the C-terminal α-helix of TTP (dark green) (pdb id code: 4J8S [67]). The side chains of the CNOT1 residues that are key to TTP binding (F847, Y851, E893, F896, Y900) are shown in red in the ball-and-stick representation; canonical hydrophobic residues characteristic of HEAT repeats are marked in yellow (same as in (**F**)). (**F**) Sequence of the CNOT1(800–999) fragment analysed in this study. The intrinsically disordered region (IDR) is followed by HEAT-like motifs; canonical hydrophobic residues characteristic of HEAT repeats are marked in yellow, other hydrophobic residues are shaded in grey. Key residues involved in TTP binding are shown in bold red. IDR fragment not visible in the crystal structure is in brackets.

The AU-rich elements (AREs) are usually repeated and/or overlapping pentamers of the AUUUA sequence [68] that are known to target mRNAs for rapid decay. They are typically located in the 3′UTR of mRNAs encoding nuclear transcription factors, proto-oncogenes, cytokines [68], and circadian clock genes [69]. Molecular recognition of the AREs of pro-inflammatory chemokine and cytokine mRNAs is mediated by tristetraprolin (TTP, also known as Nup475, G0S24, TIS11, ZPF36). TTP belongs to a family of mRNA-binding proteins that contain a cysteine-cysteine-cysteine-histidine tandem zinc finger [42]. The protein regulates mRNA stability and translation via *i.a.* the p38 MAPK or PKCα signalling pathways [70,71], which is activated by inflammatory cytokines and various environmental stresses.

TTP is thought to bind targets containing AREs in the cytoplasm following the pioneer round of translation. While the ARE-binding domain of TTP is located in the N-terminal part of the sequence [72], a short conserved helical fragment at the C-terminus, RRLPIFNRISVS, binds specifically to CNOT1 (residues 800–999) [67]. When the kinase activity is low, TTP is relatively unphosphorylated. It binds to CNOT1 and directs the CCR4-NOT deadenylase complex [41,42,73] to the target mRNA (Figure 1A), thereby triggering its 3′ deadenylation and subsequent degradation of the mRNA [41,73]. However, when the kinase pathways are activated in response to stress, TTP gets phosphorylated and is recognised by 14-3-3 adaptor proteins [74]. This event inhibits the interaction of TTP with CNOT1 [73], leading to the dissociation of CCR4-NOT from the mRNA, resulting in increased stability of the mRNA and its enhanced translation [42].

The details of the main CNOT1-TTP interaction site were investigated by in vitro co-precipitation experiments and X-ray crystallography [67]. The results showed that the conserved C-terminal amino acid sequence (314–326) of human TTP binds to the CNOT1(800–1015) fragment [67]. Disruption of this interaction by a single point mutation in the TTP, Phe319 to Ala, severely impairs, but does not completely abolish, deadenylation in vitro and mRNA decay in vivo, whereas the double mutations of Arg315 and Phe319 to Ala in TTP(174–326) abolished the interaction with CNOT1(800–1015) in vitro. The crystal structure of CNOT1(800–999) in the complex with a TTP peptide encompassing residues 312–326 has been solved at 1.5 Å resolution [67] (pdb id code: 4J8S), where residues of CNOT1(820–999) and TTP(314–325) are visible. This CNOT1 fragment consists of eight α-helices forming four HEAT repeats. The TTP peptide binds close to the N-terminus of this fragment. It inserts Leu316, Ile318, Phe319 and Ile322 into a hydrophobic groove formed between helices α1 and α3. This groove is lined with aromatic residues of CNOT1: Phe847, Tyr851, and Tyr900. The end of the groove is flanked by a negatively charged patch of amino acid residues. A salt bridge is formed between Glu893 of CNOT1 and Arg315 of TTP. Additional hydrogen bonds stabilising the interaction are formed between Glu893 and Tyr900 of CNOT1, Arg315 and Ser323 of TTP, Asn844 of CNOT1 and Pro317 of TTP, and Gln848 of CNOT1 and Arg321 of TTP. The affinity of the TTP peptide to the CNOT1(800–999) fragment was determined by isothermal titration calorimetry to be K_d_ ~2 µM [67]. The change of Phe319 to Ala in the TTP peptide weakened the interaction by 3 orders of magnitude (K_d_ ~7 mM). Thus, the interactions of these residues are crucial for the CNOT1-TTP complex formation.

Many autoimmune diseases are characterised by excessive inflammation, which can result from abnormal accumulation of pro-inflammatory chemokine and cytokine mRNAs. Elevated cytokine levels (i.e., cytokine storm) have also been observed as an indirect effect of SARS-CoV-2 infection and may increase cancer susceptibility and accelerate cancer progression [75]. They are also associated with frequent (up to 95%) reversible acute toxicities of the FDA-approved anti-cancer CAR T-cell therapy [76]. Thus, the involvement of TTP in the degradation of cytokine mRNAs may have clinical implications and has recently been considered with the potential to develop TTP-based anti-inflammatory therapies in humans [77]. Therefore, there is an urgent need for a more in-depth understanding of the biophysical basis of TTP interactions with CNOT1(800–999) in solution.

Many fragments of CNOT1 display the HEAT motif architecture. HEAT repeats, consisting of pairs of short antiparallel amphiphilic α-helices connected by a turn, were first identified in huntingtin (H), the translation elongation factor 3 (E), the regulatory A subunit of protein phosphatase 2A (A), and the signalling kinase, target of rapamycin TOR1 (T) [78]. The middle CNOT1 fragment (residues 1088–1312) consists of five HEAT-like pairs arranged into a right-handed solenoid, which is reminiscent of the MIF4G [79]. The C-terminal part is composed of two stacks of HEAT-like repeats that are perpendicular to each other [80]. It has recently been shown that the N-terminal region of CNOT1 is also folded into two helical MIF4G-like domains 1 and 2, residues 1–230 and 250–660, respectively [81]. In the CNOT1(800–999) fragment, residues 836–999 also form HEAT repeats [67]. The CNOT1(800–999) HEAT domain is composed of eight antiparallel amphiphilic α-helices belonging to five HEAT repeats, starting from the B-helix of the first HEAT motif, as shown in Figure 1F. The TTP binding site is formed by two consecutive B-helices and a turn. The hydrophobic character of the amino acid residues forming the hydrophobic core of the antiparallel helical HEAT motifs is overall well conserved [82], with the exception of one of the canonical hydrophobic residues being replaced by the glutamic acid (Glu893), which is key to the specificity of TTP binding.

HEAT repeats are structural scaffolds that, due to the versatility of this motif, can mediate interactions with other protein factors and RNAs to coordinate and regulate processes as diverse as mRNA translation initiation [83], nuclear transport, or chromosome assembly [82]. The HEAT motif, being highly prevalent, is found in numerous prominent classes of proteins, including, but not limited to, the structures of karyopherins [84,85] or the scaffolding subunit eIF4G [86,87] of the eukaryotic translation initiation complex eIF4F. However, it is worth noting that the existing scientific literature seems to contain surprisingly few results employing HDX MS in the study of α-helical proteins that feature HEAT motifs (e.g., [88,89,90]).

In this work, we show the HDX MS signature of the HEAT-like repeat domain of the human CNOT1 protein (residues 800–999) and investigate the consequences of point mutations on its structural dynamics and interactions with TTP.

## 2. Materials and Methods

### 2.1. DNA Constructs

His_6_-Sumo-CNOT1(800–999) (UniProtKB A5YKK6, isoform 2; see Appendix A) in pET 28b(+) was a kind gift from Prof. Nahum Sonenberg and Dr. Marc Fabian from McGill University, Canada. The E893A/Y900A and E893Q/Y900H point mutants were made according to QuikChange™ Site-Directed Mutagenesis (Stratagene, La Jolla, CA, USA) protocol using the primers given in Appendix A.

### 2.2. Protein Expression and Purification

Recombinant proteins were expressed in *Escherichia coli* Rosetta 2(DE3) cells. The culture was incubated at 37 °C in shaking incubator until OD600 reached ~1.0. Next, IPTG was added to a final concentration of 0.5 mM and incubation was continued O/N at 15 °C. Culture was centrifuged at 540× *g* (3000 rpm), 4 °C, for 15 min. Supernatant was discarded while pellets were either used directly for protein purification or were stored at −80 °C for purification at a later time point.

Pellets were resuspended in 20 mL of WASH buffer (50 mM sodium phosphate buffer, 300 mM NaCl, 10 mM imidazole, 2 mM DTT, pH 8.0) with 1 mM PMSF. Next, cells were lysed by sonication for 4 × 30 s on ice. After sonication, 10% Triton-X-100 was added to a final concentration of 1%. Lysate was then centrifuged at 20,000 rcf for 30 min at 4 °C. Pellet was discarded, while supernatant was used in His-affinity purification. An amount of 2 mL of slurry (beads plus solvent; HIS-Select Nickel Affinity Gel, Sigma-Aldrich, Darmstadt, Germany) per sample were placed in Bio-Rad Poly-Prep Columns and washed using 10 mL of WASH buffer. The resin was then mixed with supernatant in a 50 mL Falcon tube and incubated for 1 h in a rotator at 4 °C. Mixture was transferred to and allowed to drain through a Bio-Rad column, followed by washing 4 times with 5 mL of WASH buffer. His_6_-Sumo-CNOT1 was eluted from the resin using buffers with imidazole at 20 mM (3 mL) to get rid of impurities, 50 mM (2.5 mL) and 100 mM (2.5 mL) to elute the desired protein, and 300 mM (2.5 mL) to elute any residual protein. Protein concentration at this point was determined using Bradford method (Roti-Quant solution, Carl Roth, Karlsruhe, Germany). Fractions containing His_6_-Sumo-CNOT1(800–999) were then digested with CoolCutter protease (Genecopoeia, Rockville, MD, USA) into His_6_-Sumo and CNOT1 in 50 mM phosphate buffer, 300 mM NaCl, 50 mM imidazole at pH 8.0 for 2 h at 35 °C, using 1 U enzyme per 32 µg substrate. Buffer of digested protein samples was exchanged to 25 mM Tris/HCl, 50 mM NaCl, pH 8.0 on PD-10 desalting columns (GE Healthcare, Chicago, IL, USA). Samples were diluted twice with ultrapure water and then purified using ion-exchange chromatography on HiTrap Q HP 1 mL columns (GE Healthcare) on an AKTA FPLC. CNOT1(800–999) (pI 6.44, ProtParam [91]) was collected as the “flow-through”, while His_6_-SUMO (pI 5.96) bound to the column, if it was loaded in the diluted buffer, then it was eluted from the column at 300 mM NaCl. Freshly prepared CNOT1(800–999) was dialyzed in 50 mM Tris/HCl, 150 mM NaCl, 2 mM DTT, 1 mM EDTA, pH 7.0 using Spectra/Por 1 (Spectrum Laboratories INC, Rancho Dominguez, CA, USA) with a 6–8 kDa cut off and concentrated using Amicon Ultra 3 kDa 0.5 mL Centrifugal Filters (Merck Millipore, Burlington, MA, USA) at 4 °C on the day of HDX MS experiments. The final yield of pure CNOT1(800–999) after Ni-resin purification, cleavage and ion-exchange chromatography was approx. 5 mg/L. The molecular mass of CNOT1(800–999) after cleavage was determined by mass spectrometry to be 24,074 ± 3 Da, Appendix A (predicted mass 24,074.4 Da, ProtParam [91]).

### 2.3. Chemicals

The chemicals for protein expression and purification were purchased from Merck-Sigma-Aldrich (Darmstadt, Germany) or Carl Roth (Karlsruhe, Germany) and were analytically pure, grade A, or specified for molecular biology. The human TTP peptide APRRLPIFNRISVSE (residues 312–326, unblocked N-terminus) was ordered from Genemed Synthesis (San Antonio, TX, USA); the measured mass was 1755.0 ± 1.0 Da (average expected mass 1755.4), purity 98%; the TTP peptide concentration was determined by amino acids analysis ± 5% (Jagiellonian University, Department of Analytical Biochemistry, Kraków, Poland). The TTP concentration of 3.3 μM was chosen so that the saturation of CNOT at 2 μM was >50% (53% for K_d_ = 2 μM [67]).

### 2.4. Absorption Spectroscopy

A UV absorption spectrum of CNOT1(800–999) (Appendix A) was measured using Cary 50 from 240–320 nm to exclude contamination with nucleic acids (minimum at 247 nm) and aggregation (background at 320 nm). Protein concentration was determined based on the molar extinction coefficients at 280 nm, ∑_280_ = 23,380 M^−1^ cm^−1^ for the wt protein and ∑_280_ = 21,890 M^−1^ cm^−1^ for the E893/Y900 mutants.

### 2.5. Hydrogen–Deuterium Exchange Experiments

Each set of experiments was repeated three or four times, as previously described [20,92,93]. Each exchange and control measurement in the set was repeated at least three times. The protein was used at initial stock concentrations of 200 μM. A typical set of experiments consisted of mass spectra measurements (LC-MS runs) collected after several time intervals (10 s, 1 min, 20 min, 2 h, 6 h, and in some cases 24 h) of H/D exchange, triggered by 1:10 dilution in heavy water (99.8%, Cambridge Isotope Laboratories, Inc., Tewksbury, MA, USA) based buffer (50 mM Tris, 150 mM NaCl, 2 mM DTT, 1 mM EDTA, pD 7.0). The deuterium content in the heavy water-based buffer after addition of the chemicals was 99.27%. pD values were measured with a pH meter (pH 1000 L, pHenomenal) and were not corrected. The HDX reaction in a volume of 50 µL sample was quenched by the addition of 10 µL stop buffer (2 M glycine, 4 M guanidine hydrochloride, 150 mM NaCl, pD 2.5 in 99.8% D_2_O).

Two control measurements were carried out [92,93]. The first control experiment was related to the residual exchange that might still occur after quenching. This process is called “in-exchange”. The control was obtained by first adding stop buffer and then diluting the sample in the heavy water-based buffer to correct for a possible imperfect arrest of the H/D exchange reaction at pD 2.5. The residual exchange was then measured in the same way as in the main experiments (see below).

The experiments in the second set of controls, which measured the maximum level of the exchange possible due to deuterium loss in the process of back D/H exchange during LC-MS, were performed by in-solution pepsin digestion of protein samples under mildly denaturing conditions, followed by HDX and final MS measurements. For this purpose, immobilised porcine pepsin (Pierce) was prepared by mixing 50 μL slurry (resin plus supernatant) per back-exchange sample with a 4-fold excess of wash buffer (200 mM glycine, 0.66 M guanidinium hydrochloride, 150 mM NaCl, pD 2.5 (uncorrected), all in D_2_O), centrifuging at 1000 rcf, removing the supernatant and repeating the wash three times. Finally, the resin was mixed 1:1 with wash buffer and 50 μL of the slurry was used per sample. An amount of 5 μL of protein was diluted into 45 μL of deuterated reaction buffer, the reaction was acidified by adding 10 μL of deuterated stop buffer. The solution was incubated with the immobilised pepsin for 4 min at 4 °C with shaking. The sample was centrifuged at maximum speed in a benchtop microcentrifuge and 60 μL of the solution was transferred to a fresh Eppendorf tube. The pD of the sample was increased to ~7 with NaOD. After 2 min incubation at RT, the pD was lowered to ~2.5 by adding DCl. The sample was centrifuged again at maximum speed to remove any beads and 50 μL were injected into the nanoACQUITY UPLC system (Waters, Milford, MA, USA) and analysis continued as for normal exchange samples (see below) except that the pepsin column was removed.

This allowed us to directly correct for the back-exchange of each peptide that occurred during the chromatographic separation step performed in the H_2_O milieu. The precise back-exchange correction also allowed the effect of incomplete exchange in the 90% D_2_O buffer after dilution to be taken into account.

### 2.6. Mass Spectrometry Measurements

Immediately after HDX quenching, samples were passed through an immobilised pepsin column (Poroszyme, Applied Biosystems, Waltham, MA, USA), placed in the HDX Manager system and held at 13 °C with 0.07% formic acid (200 µL/min flow rate) as the mobile phase. Pepsin-generated peptides were then trapped on a C18 trapping column (Acquity BEH C18 VanGuard Pre-Column, Waters, Milford, MA, USA) and separated by liquid chromatography on a reversed-phase column (Acquity UPLC BEH C18 column, Waters, Milford, MA, USA) using an 8 to 40% gradient of acetonitrile in 0.1% formic acid at a flow rate of 40 µL/min using the nanoAcquity Binary Solvent Manager (Waters, Milford, MA, USA). The total run time was 13.5 min. The apparatus (except for the pepsin column) was maintained at 0.5 °C.

The peptides were then injected into the ion source of the SYNAPT G2 HDMS (Waters, Milford, MA, USA) (Q-ToF) in ion mobility mode for MS analysis. The parameters of the mass spectrometer were set as follows: ESI positive mode, capillary voltage 3 kV, sampling cone 35 V, extraction cone voltage 3 V, source temperature 80 °C, desolvation temperature 175 °C and desolvation gas flow 800 L/h. Mass spectra were acquired in the IMS mode over the *m/z* range 350 to 1500. Ion mobility separation conditions in the TriWave instrument were as follows: helium cell flow rate 180 mL/min, ion-mobility nitrogen flow rate 90 mL/min, IMS wave velocity 600 m/s and IMS wave height 40.0 V.

### 2.7. Numerical Data Analysis

The peptic peptides were identified in non-deuterated buffer on the basis of mass spectra acquired in MSE mode with the low energy function voltage of −4 eV and the high-energy function voltage of −28 eV, over the *m/z* range of 100 to 1950, in either standard or IMS mode. The isotopic envelopes after exchange were assigned to their corresponding peptide sequence tags using DynamX 2.0 or 3.0 (Waters, Milford, MA, USA), based on a peptide list obtained from the triple sequencing analysis of the unexchanged sample using ProteinLynx Global Server software (Waters, Milford, MA, USA) and a randomised database. The following acceptance criteria were used: minimum intensity threshold of 1000, minimum products per amino acid of 0.25, maximum MH+ error of 5 ppm. Identification of isotopic envelopes was performed with retention time tolerance ±0.3 min, mass tolerance ±15 ppm and drift time deviation ±2 time bins. Acquired data were verified by visual inspection and exported to Excel, R or Prism 6 (GraphPad) for further analysis.

The fraction of peptide amide hydrogens that were exchanged to deuterium, *D_f_*, for each peptide was calculated as:(1)$$D_{f}=\frac{M_{t}-{\bar{M}}_{in}}{{\bar{M}}_{bx}-{\bar{M}}_{in}}$$
where $M_{t}$ was the average mass of the peptide after a given exchange time, *t*; ${\underline{M}}_{in}$ was the average mass of the peptide in the in-exchange control samples; ${\underline{M}}_{bx}$ was the average mass of the peptide in the back-exchange control samples.

The time dependence of the Tam chodzi o cytikinenumber of hydrogens exchanged for each pepsin-generated peptide was analysed by a non-linear least-squares regression assuming for simplicity the existence of a single or dual population of amide protons of a given peptide, that exchange with the kinetics described by either a mono-exponential or bi-exponential equation, respectively:(2)$$ND\left(t\right)=\sum_{i}{ND}_{i}\left(1-e^{-k_{i}t}\right)$$
where *ND*(*t*) was the number of hydrogens exchanged to deuterium after a given exchange time, *ND_i_* were fit parameters describing the number of amide protons exchanging with the rate constants *k_i_* (min^−1^). Discrimination between mono- and bi-exponential models, and between bi-exponential models with one of the *k* values fitted or fixed [92,94,95], was based on an Akaike’s information criterion [96] or on a two-parameter Snedecor’s statistical *F*-test [97] for the models with equal or different numbers of degrees of freedom, respectively.

The intrinsic rate constant values, *k_int_*, for the studied peptic peptides were calculated according to the following numerical procedure: the peptide–amide hydrogen exchange rates for the individual amino acid pairs were predicted by the SPHERE program [98,99,100]. These values were used to calculate the expected number of deuteriums incorporated by each peptic peptide based on its sequence, and further to determine the exchanged fraction at nine different time points ranging from 0.001 to 60 s. The value of *k_int_* for each peptide was then obtained as a fitting parameter according to Equation (2).

### 2.8. Bioinformatics

The structure predictions of CNOT1(800–999) with E893A/Y900A or E893Q/Y900H point mutations was performed by AlphaFold3 [101]. The protein figures were prepared by BIOVIA Discovery Studio Visualizer v24.1.0.23298 (Dassault Systemes Biovia Corp, Waltham, MA, USA). Peptide coverage was drawn using Draw H/D Heat Map [102] or Protein Sequence Coverage Map [103]. Plots presenting D_f_ of peptides as a function of the protein sequence (Fraction exchanged in Figure 2) were generated in RStudio Version 0.99.902 using an in-house script based on Equation (1) written by Krzysztof Tarnowski and Michał Kistowski (Institute of Biochemistry and Biophysics, Polish Academy of Sciences, Warszawa, Poland). Plots of the number of hydrogens exchanged vs. time were drawn in GraphPad Prism 6.

## 3. Results

### 3.1. Structural Dynamics Properties of the CNOT1(800–999) HEAT Domain

The protection against H/D exchange is indicative of the stability of the intramolecular or intermolecular hydrogen bonding network. Therefore, it provides a description of the protein structural dynamics over a broad time scale, which is related to the frequency of transient breakages of individual hydrogen bonds and the accessibility of their protons to the solvent [104].

The first aim of our research was to characterise the HEAT motif of CNOT1 (800–999) by HDX MS to determine whether the *apo* form of the protein in solution had similar properties to those in the crystal structure in the complex with a C-terminal peptide from tristetraprolin [67]. This CNOT1 fragment is a HEAT-like bundle consisting of eight main α-helices (Figure 1E,F). In addition, some of the main helices are connected by short stretches that are also α-helical in nature. The first twenty N-terminal amino acid residues are not visible in the crystal, indicating that they are most likely unstructured.

The HDX MS experiments were performed on CNOT1 (800–999) for 10 s, 1 min, 20 min, 2 h, and 6 h of deuteration. The results were then corrected for in- and back-exchange. Peptide coverage was 99% in most experiments (at least 95%). The fractions of the proton population exchanged to deuterons are shown in Figure 2A for each pepsin-generated peptide in the CNOT1 sequence to provide a general insight into the structure and dynamics of the protein. Figure 2B shows the percentage of exchanged protons for selected peptides from this experiment as a heat map. Although the heat map simplifies the comparison of the exchange at different time points for a specific peptide, the way the data is presented in Figure 2A has the advantage of showing all the peptides measured in the experiment, which provides additional, partly redundant information based on peptides differing by a few amino acid residues.

The N-terminus (~800–840) of CNOT1(800–999) undergoes rapid deuteration, indicating that it is solvent-exposed, with main chain not involved in hydrogen bonding network, and thus most likely unstructured. In contrast, several peptides closer to the C-terminus do not undergo exchange even after 6 h. These regions must be heavily buried in the protein or be strongly involved in hydrogen bonding. When the α helices assigned to the CNOT1-TTP crystal structure were overlaid on the HDX MS results (Figure 2A,B), these strongly H/D exchange-protected regions were found to be parts of helices: α2, α3, α4, α5 and α6. However, for the vast majority of CNOT1 peptides, an intermediate extent of exchange was observed, from the fraction exchanged of 0.2 to 0.4 for the shortest time (10 s), which increased even up to 1.0 for longer deuteration times (Figure 2A). Most likely, some parts of these peptides are exposed to the solvent and become rapidly deuterated, while other parts become more deuterated only due to local fluctuations of the relatively more compact structure and transient hydrogen bond breaking during longer exchange times. Indeed, when looking at the superimposed HDX MS results (Figure 2A), it is clear that these peptides contain loops and/or the short α-helical stretches between adjacent helices that are readily accessible to the solvent. The helices α1 and α8 are poorly deuterated at the first two time points, but then undergo significant deuteration as they are at the ends of the α-helical bundle.

### 3.2. Kinetic Parameters of Hydrogen–Deuterium Exchange of the CNOT1(800–999) HEAT Domain

In Figure 2C, the deuterium uptake is plotted separately for each peptide as a function of time to reveal the kinetics of HDX. One of the simplest cases is observed for peptides 912–917, 929–936, and 946–953, which show the most slowed down HDX kinetics, and indeed they are the most buried parts of the helices: α4, α5, and α6, respectively (Figure 2D). The opposite extreme simple case is when a totally solvent-exposed peptide undergoes very rapid 100% deuteration, which is characteristic of unstructured, i.e., non-hydrogen-bonded, regions. For example, peptides 825–835 and 825–838 exchange all their amide hydrogens to deuterium rapidly, to which a mono-exponential kinetic function was fitted. The values of the number of deuterium incorporated into a given peptide (*ND*) and the exchange rate constant (k) were determined according to Equation (2). For these two peptides the results are *ND* = 9.79 ± 0.04 Da and 13.82 ± 0.04 Da, *k* = 18.7 ± 1.1 min^−1^ and 15.5 ± 0.5 min^−1^, respectively. A complete list of the kinetic parameter values for all peptides is given in Appendix A.

A more complicated case was found for the majority of the CNOT1(800–999) HEAT motif peptides showing two-step HDX kinetics, to which a two-exponential function had to be fitted according to the statistical Snedecor’s F-test or Akaike’s Information Criteria; e.g., the peptide 847–861. This is associated with the presence of at least two different classes of amide hydrogens—those that exchange more rapidly, in this case with *k* ~6–7 min^−1^, which can be seen at the first two time points in Figure 2C, and those that exchange more slowly with *k* ~0.05 min^−1^, seen at the later time points. The rates at which the amide protons of the CNOT1(800–999) HEAT motif were exchanged to deuterons could be conveniently categorised into three ranges: *k_1_* < 1 min^−1^, 1 < *k_2_* < 10 min^−1^ and *k_3_* > 10 min^−1^. The rate constant *k_1_* is associated with peptides that are relatively well protected from exchange, although not as much as those peptides for which no HDX was detected. The *k_3_* is associated with peptides that are quite easily deuterated, indicating access to solvent and/or lack of involvement in hydrogen bonding, while *k_2_* is observed for peptides that are moderately exchangeable.

The partial redundancy of the peptides allows analysis of local changes in HDX that occur at the ends of the peptides, bearing in mind that the amide hydrogen of the N-terminal amino acid residue of a peptide is invisible to HDX [99,105]. For example, in peptide 847–861 (Figure 2C), the number of incorporated deuterium determined for the hydrogens that exchange faster, with *k_2_*, is *ND* = 6.59 ± 0.19 Da, and *ND* = 4.5 ± 0.2 Da for the hydrogens that exchange more slowly with *k_1_*. On the other hand, for peptide 847–859 (Appendix A), the *ND* value for the hydrogens characterised by *k_2_* is 4.44 ± 0.17 Da, and 4.75 ± 0.19 Da for the hydrogens with *k_1_*. Thus, taking into account the differences in the *ND* values and experimental uncertainties, the two additional C-terminal residues of the longer peptide, 860 and 861 can be assigned to the faster, *k_2_*_,_ rate. These hydrogens are located at the very end of helix α2 and are thus less likely to be involved in hydrogen bonding and are more exposed to the solvent (Figure 2A,B). In the second example, the peptides encompassing residues 952–966 (Figure 2C) and 953–966 (Appendix A) have a total *ND* value of 7.38 ± 0.10 and 7.4 ± 0.2, respectively. Since the first N-terminal residue is invisible to HDX MS [99,105], any possible difference between the *ND*s of these two peptides would be associated with the residue Leu953. Since there is no difference, this residue must be protected from the exchange. This is further supported by the 946–953 peptide (Figure 2C), in which the exchange of the amide proton of this residue would be detectable. However, the whole peptide shows no detectable HDX. In the crystal structure, the Leu953 residue is located in the middle of the α6-helix (Figure 2D).

The most striking feature of the HDX signature of the CNOT1(800–999) HEAT domain is the alternating pattern of the H/D exchange-competent and exchange-protected peptide fragments, reflecting the domain structure composed of the short antiparallel amphiphilic α-helical HEAT repeats. These experiments revealed that the CNOT1(800–999) HEAT domain has a stable structure in solution, very similar to that known from crystallography, i.e., the peptides with virtually no HDX correspond to those buried deep inside the protein in the crystal structure, those with partial or delayed deuteration correspond to the fragments exposed to some extent to the solvent, while those that can exchange rapidly up to 100% correspond to the disordered fragments. These results provide a good starting point for further experiments on the TTP interaction site.

### 3.3. Protein Changes Caused by Point Mutations Monitored by Pepsin Digestion Patterns

We then investigated how point mutations designed to interfere with the TTP peptide binding might affect the structure of *apo*-CNOT1(800–999). The minor structural alterations in the folded protein that may ensue from point mutations can be examined by mass spectrometry in two ways: by analysing the pepsin digestion patterns of the protein due to possible local changes in the protein surface accessibility to the enzyme, and by measuring HDX levels for the pepsin-generated peptides. Figure 3 shows the comparison of the reproducible sequence coverage by redundant peptic CNOT1(800–999) peptides obtained for the wild type (wt) and the E893A/Y900A or E893Q/Y900H double point mutants. Interestingly, while the majority of peptides are common to both the wt protein and the mutants, there is also a significant number of novel peptides that are only detected for the mutants. The number of peptides specific for the wt protein is much lower. For both mutants, the changes in the pepsin digestion sites are mostly localised near the mutated residues, i.e., the emergence of three new sites and the loss of one site for E893A/Y900A and six new sites and the loss of one site for E893Q/Y900H. However, three new sites and the loss of three wt sites are observed for E893A/Y900A (Figure 3A) and twelve new sites and the loss of one site are observed for E893Q/Y900H in parts of the sequence distant from the mutated residues, but close in the 3D structure. These observations suggest that the local availability/susceptibility to enzymatic cleavage of some of the hydrophobic residues in CNOT1(800–999) may have been slightly altered by the introduction of the point mutations at E893 and Y900.

### 3.4. Protein Changes Caused by Point Mutations Monitored by HDX

However, the crucial question usually associated with studies based on protein point mutations is whether these mutations leave the native protein structure intact. Therefore, we performed a comparison of the HDX levels between the wt protein and its mutants to document that the changes observed in pepsin digestion patterns due to mutations do not significantly affect the secondary and tertiary structure of CNOT(800–999). Figure 4 shows the relative differences in HDX kinetics determined for the peptides that are common to the wt protein and a given mutant. The heat maps show that the differences are mainly confined to the peptide encompassing the loop containing the mutated Y900 residue, i.e., the peptide 894–911 (max. HDX diff. of 22% at 20 min) (Figure 4A) or 894–909 (max. HDX diff. of 17% at 1 min) (Figure 4B) for the E893A/Y900A and E893Q/Y900H mutants, respectively. Some smaller differences were detected for the neighbouring peptide (residues 887–893 or 888–893 for the E893A/Y900A and E893Q/Y900H mutants, respectively) containing the second mutated E893 residue belonging to the C-terminal part of the α3 helix. The HDX kinetics of the remaining CNOT1(800–999) fragments were altered to an invisible (nominally 0–5%) or negligible (5–10%) extent, which proves that the native structure of the mutant proteins has been preserved. Most importantly, the 824–845 sequence fragment, where most of the additional pepsin digestion sites for the E893Q/Y900H mutant occur, has identical HDX kinetics to the wt protein.

### 3.5. HDX MS Reveals Structural Impact of CNOT1(800–999) Mutations on TTP Binding

Having established by H/D exchange kinetics that the influence of the double point mutations on the native CNOT1 structure is negligible, we performed comparative binding studies. CNOT1(800–999) and its mutants E893A/Y900A and E893Q/Y900H, at 2 μM, were incubated with the tristetraprolin-derived peptide (TTP peptide), APRRLPIFNRISVSE (residues 312–326) at 3.3 μM, followed by HDX MS to detect the effects of TTP binding. The results are presented in Figure 5, which shows the cumulative deuteration difference (grey sticks) between the CNOT1(800–999) *apo* state and the complex with TTP. The horizontal axis shows the consecutive peptides of CNOT1, ordered according to their occurrence in the sequence. The positive values (vertical axis) indicate that peptides are becoming more protected against HDX in the complexed state. These data are semi-quantitative, having been transformed by the DynamX software, and are not corrected for in- and back-exchange, since the same sets of control peptides are not available for all CNOT1(800–999) forms. Nevertheless, these data are very useful as an indication of those regions where significant differences between the *apo* and complexed states can be observed.

TTP-binding reduces HDX levels in two regions of CNOT1 (800–999) encompassing residues 825–861 and 894–909 (Figure 5). These reduced levels are observed mainly for peptides that possess at least one of the crucial amino acid residues involved in direct TTP binding that were previously known from the crystal structure [67]: Phe847, Tyr851, Phe896, and Tyr900 (Figure 1D–F). Interestingly, an exception to this observation is the peptide no. 39, PDKELHITA (residues 901–909), which does not contain any of the above residues but for which a significant difference in HDX was observed. This means that the effect of TTP binding on the protection of the CNOT1(800–999) amide hydrogens reaches further, since HDX can reflect the rearrangement of hydrogen bonds or changes in solvent accessibility not only for strict binding sites, but also in their vicinity. Figure 6A presents all the CNOT1 residues that are within a distance of 3.5 Å to the TTP peptide atoms in the crystal structure [67]. Two of them, Pro901 and Glu904, belong to the 901–909 peptide. Although a proline residue does not have an amide hydrogen, the proximity of this CNOT1 region to TTP can influence the HDX level of the surrounding residues, particularly since the Cδ atom of the Pro901 ring is involved in a non-classical carbon hydrogen bond with the backbone oxygen atom of Ile322 of TTP with a length of 3.18 Å. In Figure 6B, these two regions of the CNOT1 structure have been highlighted in red, with the intensity of the red colour corresponding to the HDX protective effect.

The introduction of point mutations into the TTP binding site of CNOT1 results in a loss of affinity for the TTP peptide (Figure 5). However, the severity of the effect is slightly different depending on whether the functional groups of the key residues have been changed to the neutral alanine side chains or to such groups that retain a partially electrostatic or ring character. While the E893A/Y900A mutations completely abolish the TTP binding (Figure 5 and Figure 6C), some residual evidence of the interaction with TTP could be detected for the E893Q/Y900H mutant in the region of the intrinsically disordered N-terminal part of CNOT1(800–999) and the beginning of its α1 helix (Figure 5 and Figure 6D).

## 4. Discussion

### 4.1. HDX Signature of the CNOT1(800–999) HEAT Domain

Although the HEAT repeats are structural scaffolds of many eukaryotic proteins that are involved in regulation of a variety of cellular processes, only a small number of HDX MS results are available for proteins containing HEAT motifs (e.g., [88,89,90]). In particular, there are no HDX MS data related to CNOT1, while many fragments of this large protein display the HEAT repeat architecture. The CNOT1(800–999) fragment analysed in this study is composed of three full HEAT repeats, flanked by one N-terminal B-helix, and one C-terminal A-helix of two more HEAT repeats [67]. The HDX MS signature of the CNOT1(800–999) HEAT domain features an alternating pattern of H/D exchange-competent and exchange-protected peptide fragments, corresponding to the presence of short antiparallel amphiphilic α-helices. In particular, the HDX kinetics of the peptic CNOT1(800–999) peptides exhibit two-exponential behaviour (see Appendix A), indicative of the presence of at least two populations of amide hydrogens that exchange rapidly, moderately fast, or slowly. This observation can be attributed to the fact that peptic peptides typically contain both loops and compact α-helical fragments. HDX MS analysis detected also the fragments that are deeply buried inside the HEAT domain.

### 4.2. Control of the Protein Fold Versus Point Mutations

In general, when designing functional studies based on protein point mutations, it is essential to ensure that these mutations leave the protein structure intact and that the observed effects can be unambiguously attributed to changes in the functional groups of the mutated residues. Pepsin is thought to cleave proteins non-specifically, the only determinant being the presence of a hydrophobic residue [106]. However, there is evidence that for a given protein under the same conditions, this is reproducible, resulting in a constant pattern of peptic peptides from that protein [107], which suggests that the local protein conformation may also play a role by regulating the accessibility of the hydrophobic residues to be cleaved. In our previous work on conformational changes of the eukaryotic translation initiation 4E factor (eIF4E) due to mRNA 5′ cap binding, we also noticed that the transition from the *apo* state to the ligand-bound state is accompanied by a slightly altered pepsin digestion pattern [92]. In the case of CNOT1(800–999), we show that the analysis of the changes in the peptic peptides obtained after the introduction of point mutations can be used as a sensitive tool to control the correct protein folding.

In parallel, the HDX approach makes it possible to reveal any structural changes that may occur as a result of point mutations. The HDX differences found locally for the mutated CNOT1(800–999) sites are not surprising. However, an important observation is that HDX MS is sensitive enough to detect both very small changes in parts of the protein away from the mutation sites and to demonstrate that they are negligible in the context of the binding studies. This provides the evidence needed to infer from point mutations that it is the intended binding site and not a perturbation in the whole protein structure that is responsible for the loss of the interaction.

### 4.3. TTP Binding Site of CNOT1(800–999) in Solution

In the crystal structure [67], the TPP binding site of CNOT1(800–999) is composed of the C-terminal parts of the α1-helix (B-helix in the HEAT repeat) with Phe847 and Tyr851, and the α3-helix (also B) with Glu893, and two residues Phe896 and Tyr900 belonging to the turn between α3 and α4 helices. The TTP peptide is partially inserted into a hydrophobic groove of CNOT1(800–999) lined with amino acids: Phe847, Tyr851 and Tyr900. The TTP peptide also forms a salt bridge with Glu893 and hydrogen bonds with Asn844, Gln848, Glu893 and Tyr900 (Figure 6A). Contrary to typical HEAT repeats, one of the canonical hydrophobic residues of the B-helix is replaced by the Glu893 (Figure 1F).

A more quantitative comparison of the HDX, corrected for back-exchange and taking into account the lack of information for prolines and N-terminal residues, obtained for the available partially redundant CNOT1(800–999) peptides for the wt protein in the *apo* state and in the presence of TTP allowed us to calculate the differences in deuterium uptake for given residues. The results of this analysis are summarised in Table 1. Since the TTP-CNOT1(800–999) interaction is relatively weak (K_d_ ~2 μM [67]), the net Δ*ND* values are moderate at this protein stoichiometry in solution.

Importantly, in addition to the changes observed for the peptides containing the residues known from the crystal structure [67] to be key for TTP binding, significant HDX changes were also found for the DDEANSY(840–846) sequence and for the DKE(902–904) and LHITA(905–909) fragments. These changes suggest the involvement of a larger part of CNOT1(800–999) in TTP recognition, which may be more deeply buried or conformationally affected upon TTP binding. The HDX MS measurements of the interactions of CNOT1(800–999) with TTP in solution have, therefore, shown that the region stabilised by TTP is extended beyond the classic binding site, at least from residues 840 to 859 and from 895 to 909, and that the DDEANSY(840–846) sequence preceding residue Phe847 is also involved in TTP binding.

For the double point mutations at E893 and Y900, the HDX changes observed for wt CNOT1 due to TTP binding are completely lost (Figure 5), indicating that the ability of the mutants to bind TTP is severely compromised. In principle, the binding mode could be altered in the mutant protein, leaving some residual affinity with a different binding pattern, which cannot be excluded and is most likely what we observe for the EY/QH mutations (Figure 5 and Figure 6D). Importantly, however, the E893/Y900 mutations completely abolish these specific HDX changes that are observed for typical binding to the wt protein in the region of 839–861 and 895–909 (Figure 5 and Figure 6A, Table 1), implying that the mutated sites play a significant role in this interaction.

These results of the studies in solution suggest that the role of the Glu893 and Tyr900 functional groups is dominant in the formation of the specific network of inter- and intramolecular non-covalent contacts involving the crucial Arg315 residue of TTP that stabilise the complex with CNOT1(800–999), i.e., the salt bridge and hydrogen bond with Glu893, the cation-π stacking with Phe896, and the intramolecular hydrogen bond between Glu893 and Tyr900. The other residues can support the complex formation to a lesser extent.

The HDX MS studies of CNOT1(800–999) demonstrate that this methodological approach is a powerful tool, providing valuable information that is difficult to obtain by other methods for proteins that are available at limited concentrations, precluding, e.g., multidimensional NMR measurements that could yield atomic resolution. The advantages of HDX MS over NMR have recently been shown for the encephalitis virus capsid protein [29]. On the other hand, HDX MS results can provide information with a fairly precise resolution of a few, sometimes even single amino acid residues, in contrast to circular dichroism, which is commonly used to check for the absence of perturbations in protein structure caused by point mutations, where only the total content of secondary structure elements can be determined [108].

Since CNOT1 may be involved in the formation of molecular condensates, such as P-bodies, through interactions with the GW182 protein [44], more complex systems are emerging as possible future targets for HDX MS studies in the context of liquid–liquid phase separation (LLPS). Recently, HDX MS has been shown to be able to probe protein structure and conformational changes directly in condensates, e.g., in the case of LLPS involving Pab1, HDX MS allowed the identification of interaction networks that drive condensation [10].

## 5. Conclusions

HDX MS is one of the few methods that can control the conformational consequences of introducing point mutations with relatively high precision. The absence of non-specific changes caused by such mutations is a prerequisite for further functional studies of protein–protein interactions. In the case of TTP binding by CNOT1(800–999) in solution, HDX MS measurements have shown that the Glu893 and Tyr900 residues of CNOT1 are crucial for this interaction, but the complex formation is also accompanied by changes in the hydrogen bonding network involving amide hydrogens from the regions DDEANSY(840–846), DKE(902–904) and LHITA(905–909), which point to involvement a larger part of CNOT1 in the molecular recognition of TTP. On the other hand, while the HEAT repeat is one of the most common canonical protein motifs, our results are, surprisingly, one of the rare HDX MS analyses that address the properties and interactions of HEAT-like domains. Taken together, the results demonstrate the power of HDX MS in elucidating the sub-molecular basis of protein functions.

## Figures and Tables

**Figure 2 biomolecules-15-00403-f002:**
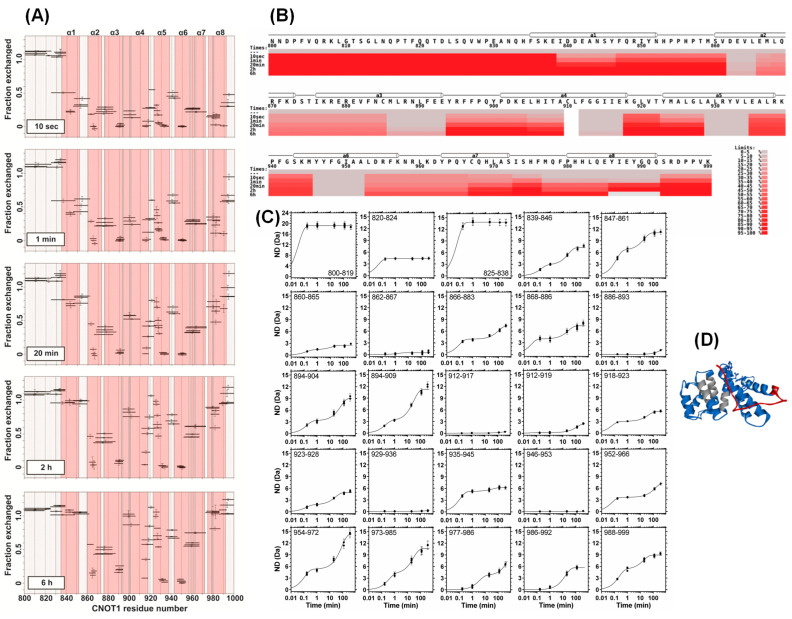
(**A**) HDX MS signature of the HEAT repeat domain of the human CNOT1(800–999). The fraction of protons’ population exchanged to deuterium (Equation (1)), of pepsin-generated peptides of CNOT1(800–999) (horizontal lines) after 10 s, 1 min, 20 min, 2 h and 6 h of deuteration. Vertical lines show the standard deviations from three repeats of the experiment. The α-helical regions are marked in pink. The fractions exchanged of apparent values of > 1.0 are the result of the imperfect nature of the back-exchange correction, which nevertheless is indispensable to compare the HDX of different peptides between each other. (**B**) Heat map of CNOT1(800–999) HDX. Fraction exchanged (%) after different deuteration times (starting from top: 0, 10 s, 1 min, 20 min, 2 h, 6 h) for selected, non-redundant pepsin-generated peptides of CNOT1(800–999) is displayed in colors: ranging from grey (0–5%) up to red (95–100%). (**C**) Example HDX kinetics of subsequent, partially redundant CNOT1(800–999) peptides. Deuterium uptake of the peptides is shown as a function of time. Data corrected for back-exchange. (**D**) Crystal structure of CNOT1(800–999). Peptides of immediate HDX are marked in red, peptides of complex HDX kinetics are marked in blue, and exchange-inaccessible peptides are marked in grey. The side chains of the residues that are key to TTP binding (F847, Y851, E893, F896, Y900) are shown in the ball-and-stick representation.

**Figure 3 biomolecules-15-00403-f003:**
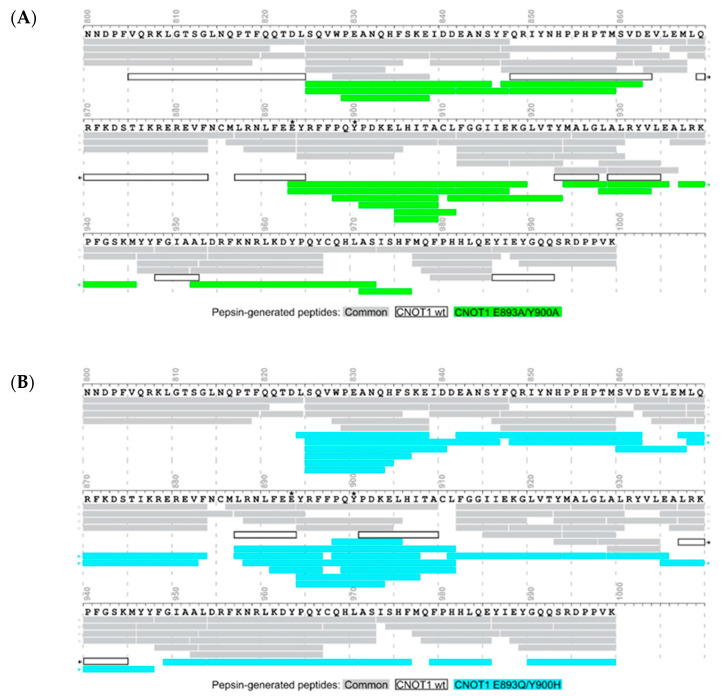
Reproducible sequence coverage by redundant peptides obtained by pepsin digestion for CNOT1(800–999) with double point mutations, (**A**) E893A/Y900A and (**B**) E893Q/Y900H, with respect to the wild-type protein. Grey bars—common peptides for the wild type and mutant proteins; white bars with black border—peptides specific for wt CNOT1(800–999), green and cyan bars—peptides specific for CNOT1(800–999) E893A/Y900A and E893Q/Y900H mutants, respectively.

**Figure 4 biomolecules-15-00403-f004:**
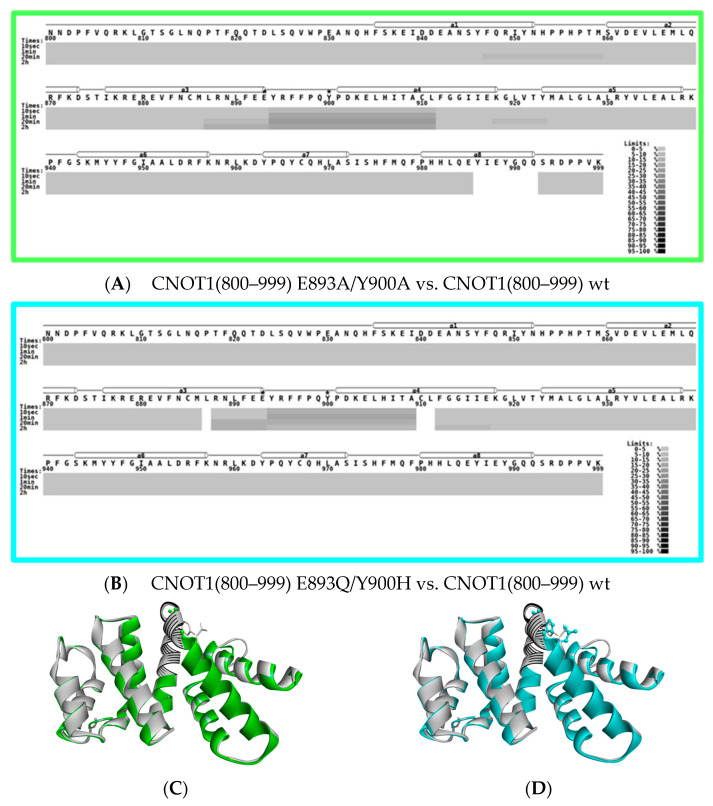
Heat maps of relative differences in HDX kinetics for CNOT1(800–999) with double point mutations, (**A**) E893A/Y900A and (**B**) E893Q/Y900H, with respect to the wild-type protein. The mutated residues, E893 and Y900, are marked with asterisks. Data corrected for back-exchange. (**C**) Three-dimensional structure generated by AlphaFold3 [102] for CNOT1(800–999) E893A/Y900A (green) and (**D**) CNOT1(800–999) E893Q/Y900H (cyan) superimposed on the crystal structure of wild-type CNOT1(800–999) (pdb id code: 4J8S [67], grey). The peptide fragments showing differences in HDX are marked as black line ribbons.

**Figure 5 biomolecules-15-00403-f005:**
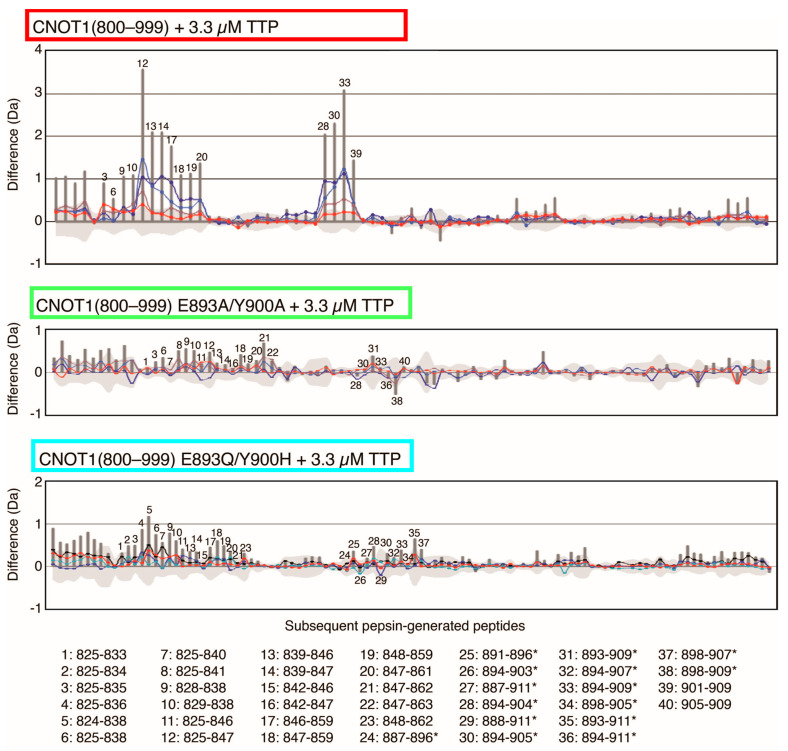
Difference plots for HDX of CNOT1(800–999) and its double point mutants, E893A/Y900A and E893Q/Y900H, each at a concentration of 2 μM in the presence of TTP peptide at 3.3 μM after different deuteration times: 10 s (red), 1 min (brown), 20 min (blue), 2 h (dark blue); grey vertical bars—cumulative difference; grey shadow—experimental uncertainty. Partially redundant pepsin-generated CNOT1 peptides are numbered in the graphs, and the ranges of CNOT1 residues belonging to specific peptides are given below the graphs. Sequence of CNOT1(800–999) is shown in Figure 1D. Data uncorrected for back-exchange. Peptides containing the point mutations are marked by asterisks (*).

**Figure 6 biomolecules-15-00403-f006:**

Visualisation of the HDX differences due to the TTP peptide binding. (**A**) CNOT1(800–999) structure (blue) in the complex with the TTP peptide (dark green) (pdb id code: 4J8S [67]); the side chains of the CNOT1(800–999) residues in contact with TTP marked in red in the ball-and-stick representation. (**B**) CNOT1(800–999) fragments that display stronger (red) and weaker (pink) HDX differences in the presence of TTP; the residues key to TTP binding (F847, Y851, E893, F896, Y900) are shown in CPK colours in the ball-and-stick representation. (**C**) CNOT1(800–999) E893A/Y900A with the mutated residues marked in green in stick representation. (**D**) CNOT1(800–999) E893Q/Y900H with the mutated residues marked in cyan in stick representation; protein fragments that display negligible HDX differences in the presence of TTP are marked in light cyan.

**Table 1 biomolecules-15-00403-t001:** Differences in total HDX levels of the CNOT1 fragments between the *apo* state and in the presence of the TTP peptide after 2 h of deuteration. Δ*ND*, difference in the number of deuteriums incorporated by a given fragment in Da; CNOT1 at 2 μM, TTP at 3.3 μM; data corrected for back-exchange.

CNOT1						
Res. no.	840–846	847	849–859	895–901	902–904	905–909
Sequence	DDEANSY	F	RIYNHPPHPTM	RFFPQYP	DKE	LHITA
Δ*ND* (Da)	1.6	1	1.5	1.4	0.5	0.4

## Data Availability

The files containing *m/z* distributions for all quantifiable peptides of acceptable quality are deposited in the RepOD Open Research Data Repository (https://doi.org/10.18150/IEVM0L accessed on 5 February 2025).

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
