# Peer review of "Exploring the CNOT1(800–999) HEAT Domain and Its Interactions with Tristetraprolin (TTP) as Revealed by Hydrogen/Deuterium Exchange Mass Spectrometry"

_biomolecules, 2025, doi:10.3390/biom15030403_

Round 1
Reviewer 1 Report
Comments and Suggestions for Authors
The manuscript by Cieplak-Rotowska et al. illustrates a detailed H-D exchange study carried out with mass spectrometry. The work done by the authors is undoubtedly remarkable, whereas the results do not seem to justify the amount of effort that was spent. The manuscript presents a very long and complex introduction that can be shortened substantially as most of the described particulars are reported elsewhere (the authors mention several references). The introduction could address directly the complex of interest which does not require the entire framework described. In addition the authors should explain better the statement reported in lines 56-65. An additional explanation is necessary for the overexpression risks (line 70).
The manuscript would also benefit from a reduction of the result section to make quicker the reading and reach more effectively the salient discussion points.
With these correction, I think the work could be published.
Reviewer 2 Report
Comments and Suggestions for Authors
Cieplak-Rotowska, et al., present an interesting and thorough HDX MS study of a HEAT domain from CNOT1 that plays a critical role in RNA regulation. They compare the wild type of this domain to two double-point mutants in the absence and presence of a peptide representing one of CNOT1's key functional binding partners, TTP.
Overall, this study is well-executed with high-quality data that is clearly presented. The study demonstrates both the power of HDX-MS in allowing comparison of structure and dynamics for point mutants and in yielding new insight on the protein system studied. The data convincingly show that the functionally deleterious point mutants do not create large-scale changes in structure or dynamics and that the wild type protein includes regions of varying dynamics including, quite interestingly, some regions that exhibit bimodal dynamics on markedly different time frames that are resolved by the authors' data. The authors also successfully identify a previously unknown region of the HEAT domain that is influenced by TPP binding.
Despite the excellent presentation of the data and powerful conclusions related to the impact of point mutation, I struggle with the authors' interpretation of the peptide-binding data. The authors mention that the mutations tested have been shown to severely affect TTP binding (lines 814-815), but there is no citation associated with this assertion, nor do the authors state how severely the peptide affinity is altered by these mutations. In their interpretation of their own data, wherein 1.6-fold excess peptide was included, they conclude that the point mutants failed to bind the peptide. This conclusion seems to be drawn from the apparent loss of binding-dependent changes in protection factors, but I don't understand how the authors reach this conclusion based on the information presented. The design of the peptide-binding portion of this study seems to be aimed at investigating the effect of the point mutants on the structural and dynamical effects of TPP binding, yet the authors seem to conclude that their point mutant experiments in the presence of peptide were not actually peptide-bound? If that is the case, what is the value of these data? Do they not merely serve as a reproduction of the apo measurements? Why not perform the measurements with a large enough excess of peptide to drive the point mutants to a peptide-bound state?
When I studied the results presented in Figures 5 and 6, I interpreted them as showing that the point mutants abrogated the binding-induced changes in protein dynamics seen for the wild type. I was stunned and excited by this result, fascinated by the idea that the authors had identified key residues linked to dynamical aspects of binding thermodynamics. Instead, the authors concluded that the absence of those changes meant the peptide did not bind. I do not see how the data presented can be interpreted as representative of binding or not binding. Surely, evidence of the fraction of HEAT domain that is TPP-bound must come from some other more direct measure. If the authors knew that the affinity was so low that the peptide would not bind to the mutants, why not include a large enough excess to drive peptide binding? This aspect of the study and the interpretation of the data needs to be improved/clarified prior to publication.
Round 2
Reviewer 2 Report
Comments and Suggestions for Authors
The edits to the manuscript greatly improve clarity, and the authors have addressed all of my concerns.
Author Response
Comment1: The edits to the manuscript greatly improve clarity, and the authors have addressed all of my concerns.
Response1: Thank you for the positive comment.